# Activation of L-histidine biosynthesis as a new antibiotic strategy against *Mycobacterium tuberculosis*

Debbie M. Hunt[1,8,9], João Pedro Pisco [1,7,8], Angela Rodgers[2], Cesira de Chiara [1], Anisha Zaveri[3], Kamila L. Pacholarz[4], Dimitrios Evangelopoulos [5], Acely Garza-Garcia[1], Sabine Ehrt [3], Dirk Schnappinger [3], Perdita E. Barran [4], Maximiliano G. Gutierrez [2] & Luiz Pedro S. de Carvalho [1,6] ✉

The increasing prevalence of antimicrobial resistance is an important challenge that warrants new approaches to antibiotic development. Currently, all antibiotics inhibit biological processes. To explore whether activation of a biochemical pathway can elicit bactericidal effects we engineered variants of *Mycobacterium tuberculosis* ATP-phosphoribosyltransferase (ATP-PRT) that are resistant to allosteric inhibition by L-histidine, leading to supraphysiological activation of ATP-PRT and L-histidine overproduction. Upregulation of L-histidine biosynthesis significantly reduces the growth of *M. tuberculosis* in culture and causes a loss of fitness owing to nutrient and energy depletion. Moreover, the expression of allosteric variants in *M. tuberculosis* significantly reduced infections in human macrophages and in a mouse model of infection. Thus, metabolic activation represents a new mycobactericidal mechanism that could be applied to antimycobacterial drug discovery.

New antibiotics with innovative mechanisms of action (MoA) are urgently needed to reverse the global rise in antimicrobial resistance, or AMR[1–6]. If this goal is not met by 2050, it has been estimated that 10 million people will die every year from previously treatable infections[7]. For tuberculosis, caused by the Gram-positive bacterium *Mycobacterium tuberculosis*, WHO estimates that more than 400,000 people developed multi-drug resistance or rifampin resistant TB in 2023[8]. The development of antibiotics that can overcome AMR has been thwarted by a combination of high failure rate (95%), elevated cost (hundreds of millions of dollars per molecule), and limited profitability[9], leading to an exodus of pharmaceutical companies from discovery and development in this therapeutic area. Meanwhile, academic laboratories have generated several candidates and a few clinically approved antibiotics with new MoA centered on inhibiting essential processes. For

example, bedaquiline is a mycobacteria-specific ATP synthase inhibitor that was approved by the US Food and Drug Administration (FDA) in 2012 for the treatment of drug-resistant tuberculosis[10]. In addition, inhibitors of lipopolysaccharide transport machinery are active against Gram-negative bacteria[11,12] and inhibitors of lipoprotein transport are bactericidal to Gram-negative bacteria[13]. Moreover, all antibiotics in clinical use are inhibitors. For example, rifamycins inhibit RNA polymerase[14], oxazolidinones, tetracyclines and aminoglycosides inhibit the ribosome[15–17], penicillins and cephalosporins inhibit peptidoglycan transpeptidases and therefore block cell-wall biosynthesis.

Human pharmacology includes numerous examples of receptor agonizts and enzyme activators[18]. For example, morphine and codeine are opioid receptor agonizts, semaglutide is a GLP-1 receptor agonist, plecanatide is an activator of soluble guanylate

[1]Mycobacterial Metabolism and Antibiotic Research Laboratory, The Francis Crick Institute, London, United Kingdom. [2]Host-Pathogen Interactions in Tuberculosis Laboratory, The Francis Crick Institute, London, United Kingdom. [3]Department of Microbiology and Immunology, Weill Cornell Medicine, New York, United States of America. [4]Manchester Institute of Biotechnology, The University of Manchester, Manchester, United Kingdom. [5]Department of Microbial Diseases, UCL Eastman Dental Institute, London, United Kingdom. [6]Department of Chemistry, The Herbert Wertheim UF Scripps Institute for Biomedical Innovation & Technology, Jupiter, United States of America. [7]Present address: LifeArc, Accelerator Building, Open Innovation Campus, Stevenage, United Kingdom. [8]These authors contributed equally: Debbie M. Hunt, João Pedro Pisco. [9]Deceased: Debbie M. Hunt. ✉e-mail: soriodecarval.lp@ufl.edu

cyclase C, and mitapivat is a pyruvate kinase activator. Indeed, pharmacologists have identified clinically useful super-agonists, such as goserelin, which binds to the gonadotropin-releasing hormone receptor, suppressing the production of sex hormones. We hypothesized that metabolic activation, in which a pathway is kept ON, or its rate is increased beyond physiological requirements, could represent a new avenue of antibiotic discovery. Activation of metabolic pathways can promote cellular toxicity through at least three different mechanisms: (i) depletion of essential metabolic precursors, (ii) energy depletion (*e.g.*, ATP depletion) or (iii) accumulation of toxic intermediates and/or end products. Importantly, activation and agonism are distinct from overexpression of a target, as they work with native, physiological levels of the target protein. While activation and agonism can be straightforwardly generated by a small molecule, specific overexpression of a gene by a small molecule is different and only possible when the levels of the protein in question are not tightly regulated.

L-histidine is an essential amino acid required for protein synthesis in all living organisms. Thus, it must be obtained through diet or biosynthesis. L-histidine biosynthesis is present in bacteria, fungi and plants, but absent in humans. In addition, L-histidine biosynthesis is connected with purine biosynthesis via the common intermediate 5-aminoimidazole-4-carboxamide ribotide (AICAR)[19]. This couples L-histidine biosynthesis with purine metabolism and impacts a multitude of biological processes beyond these two pathways, such as coenzyme A biosynthesis[20]. L-histidine is also a precursor of ergothioneine (EGT)[21], an essential thiol-containing one-electron small molecule antioxidant. In addition, L-histidine biosynthesis is highly energetically demanding, consuming a total of 41 equivalents of ATP per molecule made[22]. It has been shown that L-histidine is not present at sufficient levels in host niches occupied by *M. tuberculosis*, as illustrated by the essentiality of all *his* biosynthetic genes in vitro and in vivo[23–26]. Furthermore, CRISPRi of L-histidine biosynthesis genes in *Mycobacterium smegmatis*, a fast-growing saprophyte often used as a model of *M. tuberculosis*, led to a clear filamentation phenotype, a morphological change that indicates defective cell division[27]. Moreover, inactivation of *hisD* leads to an interferon-γ-dependent attenuation of infection in mice, indicating that *M. tuberculosis* relies on de novo L-histidine biosynthesis to counter interferon-γ triggered immune activation[28]. Therefore, disruption of L-histidine biosynthesis may have poly-pharmacological effects in *M. tuberculosis*, affecting metabolism, cellular ultrastructure, and ultimately bacterial viability during infection, making this pathway a privileged target for antibiotic discovery.

L-histidine produced de novo or taken up by cells induces feedback inhibition of ATP phosphoribosyltransferase (ATP-PRT - EC 2.4.2.17), the first enzyme of the L-histidine biosynthesis pathway (Supplementary Fig. S1A). *M. tuberculosis* ATP-PRT is hexameric, and it catalyzes the formation of phosphoribosyl-ATP from ATP and phosphoribosylpyrophosphate (PRPP; Fig. 1A), the first and committed step in L-histidine biosynthesis. For the *M. tuberculosis* enzyme, L-histidine is an uncompetitive inhibitor versus ATP ($K_{ii\text{-}ATP} = 28 \pm 2 \, \mu M$) and a non-competitive inhibitor versus phosphoribosyl-pyrophosphate (PRPP) ($K_{ii\text{-}PRPP} = 23 \pm 6 \, \mu M$, $K_{is\text{-}PRPP} = 26 \pm 12 \, \mu M$)[29,30]. L-histidine binding to the allosteric domain of *M. tuberculosis* ATP-PRT leads to stabilization of the closed form of the hexamer, due to reduction of the frequency of open to close transition (Fig. 1B) and complete inhibition of the reaction[29–31]. Importantly, *M. tuberculosis* ATP-PRT can be activated in vitro, even in the presence of saturating levels of L-histidine, by a thiophene-L-alanine small molecule activator[30], indicating that this enzyme can be activated and allosteric inhibition can be overcome. The current lack of potent and selective small-molecule activators of ATP-PRT with whole-cell activity limits the pharmacological testing of the hypothesis that disrupting allosteric feedback inhibition would be bactericidal. Nonetheless, L-histidine biosynthesis remains an attractive target for antibiotic discovery

and an interesting opportunity to test whether engineered activation, leading to L-histidine overproduction, can overcome the natural feedback inhibition and ultimately kill bacteria.

In this study, we demonstrate the potential of metabolic activation as an antibiotic strategy. We designed and characterized ATP-PRT allosteric variants that are insensitive to inhibition and could even be activated by L-histidine binding. This led to overactivation of the entire pathway and L-histidine overproduction, which resulted in a growth defect in vitro. The loss of viability significantly limited *M. tuberculosis* infection of human macrophages and displays bactericidal activity in a mouse model using low-dose aerosol infection.

## Results

### Natural allosteric inhibition as a chassis for engineering activation

To identify allosteric variants of ATP-PRT that are insensitive to L-histidine inhibition or activated by it, we analyzed allosteric site residue conservation using multiple sequence alignments, phylogenetics, and the X-ray structure of *M. tuberculosis* ATP-PRT bound to L-histidine[30]. Primary sequences of 20 homo-hexameric ATP-PRT enzymes from bacteria, fungi and plants were used to build a structure-guided multiple sequence alignment (Supplementary Fig. S1B, S1C). This alignment revealed residues in the allosteric site that are fully conserved, such as T238 (likely essential for the fold or for allostery), residues that are highly conserved, such as L275, and residues that are poorly conserved, such as D216 (Fig. 1C, D). Based on interaction with L-histidine we selected the following residues D216, D218, T238′, A249, L253, A273 and L275 and designed mutations to potentially alter allosteric regulation by changing how L-histidine interacts with its binding site, while hoping to minimize structurally incompatible modifications. Eight variants were constructed, recombinantly expressed in *Escherichia coli*, purified, and confirmed by DNA sequencing and by intact protein mass spectrometry (Supplementary Fig. S2A, S2B).

The eight ATP-PRT variants were then analyzed by circular dichroism (CD) spectroscopy. CD indicates that seven out of eight variants were correctly folded, illustrated by CD profiles identical to the one obtained with wild-type ATP-PRT (Supplementary Fig. S3). The ATP-PRT L275D variant was excluded from subsequent experiments as it is partially misfolded or unfolded (Supplementary Fig. S3). We next evaluated thermal unfolding ($f_U$) as an indication of functional conformational changes, in the absence and presence of L-histidine, compared with wild-type ATP-PRT. Closed (inhibited) forms of ATP-PRT unfold at higher temperatures (*i.e.*, they are more stable) than open forms, as seen for free and ligand-bound wild-type ATP-PRT (Fig. 1E). Notably, binding of L-histidine to ATP-PRT leads to a 13 °C increase in unfolding temperature ($T_m$), consistent with a substantial stabilization of the closed form of the hexamer. Unfolding behavior of some ATP-PRT variants deviated from the wild-type profile (Fig. 1E), and could be divided into three groups, based on the overall results: (i) D218L, A273G and L275A display unfolding profiles that are very similar to wild-type ATP-PRT (*i.e.*, L-histidine binding leads to the same increase in stability observed with the wild-type enzyme); (ii) L253′A did not display increased stability upon L-histidine binding, indicating either that L-histidine does not bind or that ligand binding does not trigger the conformational change leading to the closed form, yet it appears to be already closed. Consistent with this interpretation, this mutant was inactive and not investigated further; and (iii) D216V, T238′V and A249K display unfolding profiles that are inconsistent with L-histidine binding or they fail to produce the conformational changes leading to a more stable and closed hexamer. Together, these results demonstrate that mutations at the L-histidine binding site can lead to stable ATP-PRT allosteric variants that differentially bind to or respond to L-histidine binding.

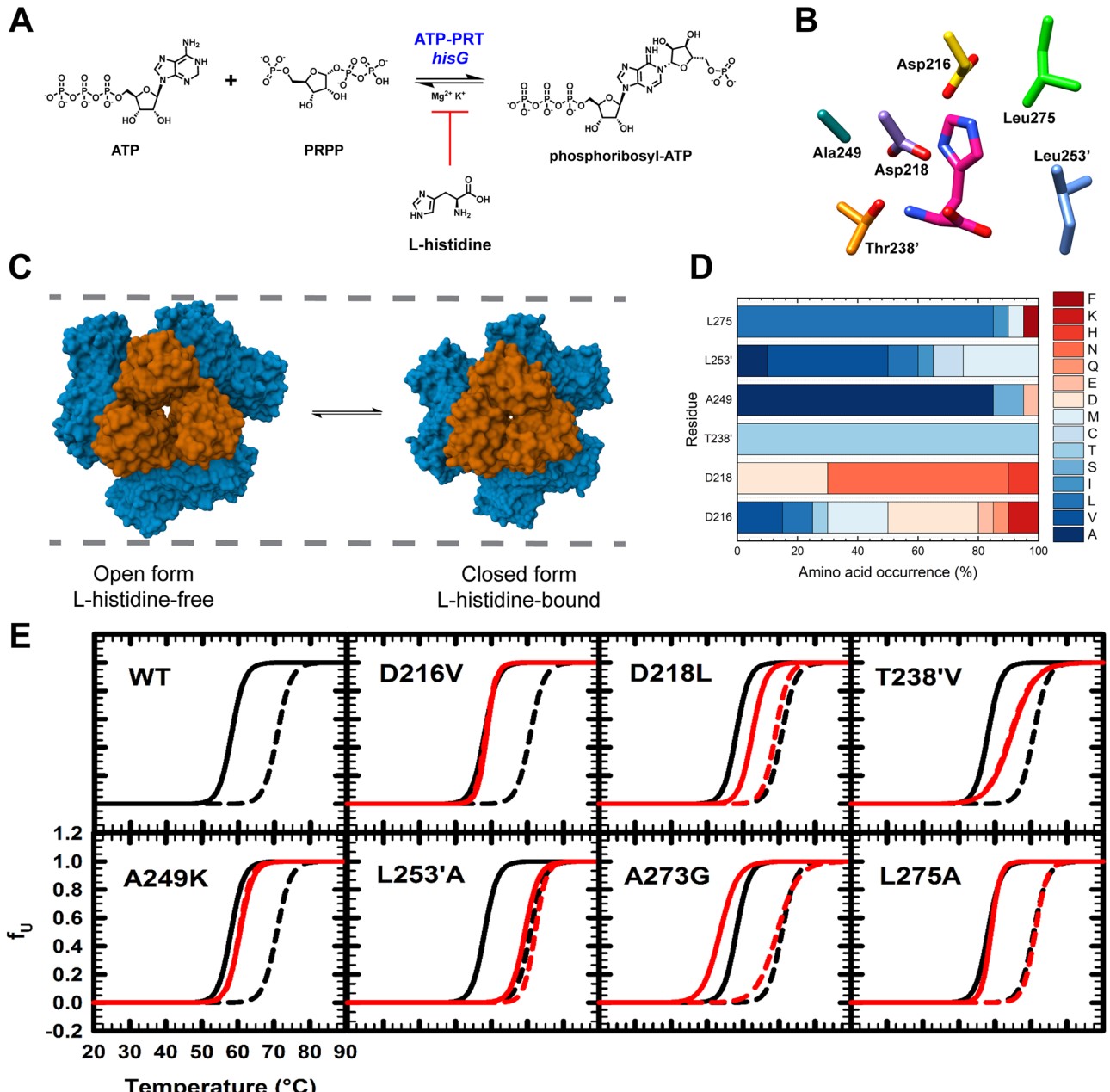

**Fig. 1 | Generation of allosteric variants of ATP-PRT. A** Overall reaction catalyzed by ATP-PRT. **B** Scheme illustrating the 3-dimensional structure of *M. tuberculosis* ATP-PRT in open and closed (L-histidine bound) conformations (PDB ID 1NH7 and 5LHU, respectively). The catalytic domain is shown in blue, and the allosteric domain is in gold. **C** Scheme of the allosteric site of *M. tuberculosis* ATP-PRT with L-histidine bound, illustrating the residues chosen for mutagenesis. **D** Amino acid occurrence at the six positions identified in panel (**C**). **E** Thermodynamic stability (expressed as the fraction of enzyme unfolded - $f_U$) of ATP-PRT wild-type (black traces) and variants (red traces) in the absence (solid line) and presence (dashed) of L-histidine. Lines represent the fit of the data. The data are representative of two independent experiments.

## Allosteric variants alter steady-state kinetics and allosteric regulation by L-histidine

To demonstrate whether these ATP-PRT variants display altered kinetics and allosteric regulation, we determined steady-state kinetic parameters in the absence and presence of L-histidine, using a coupled enzymatic system[29]. Figure 2 shows data obtained for wild-type ATP-PRT and for the D216V, D218L, L253'A, T238'V, A249K and A273G variants. $V_{max}$ ($k_{cat}$) and $K_m$ for ATP were minimally altered in these mutants. By contrast, nearly all variants display changes in the $K_m$ for PRPP, reaching up to 20-fold higher (lower affinity), for example, with the T238'V variant (Table 1). Importantly, steady-state kinetics reveal that ATP-PRT variants D216V,

T238'V and A249K are no longer inhibited by L-histidine, even at concentrations approaching 1000-fold higher than the $K_i$ for the wild-type ATP-PRT (Fig. 2). Surprisingly, the D216V and A249K variants are also activated by L-histidine, revealing a complete reversal of the native allosteric regulation, with a single residue substitution. The lack of inhibition and even activation by L-histidine observed with ATP-PRT D216V, T238'V and A249K variants correlates with the lack of thermal stabilization, determined in thermal unfolding experiments (Fig. 1E). Therefore, these variants represent unique reagents to interrogate the impact of overriding negative allosteric regulatory networks in bacterial metabolism, fitness, and in pathogenesis.

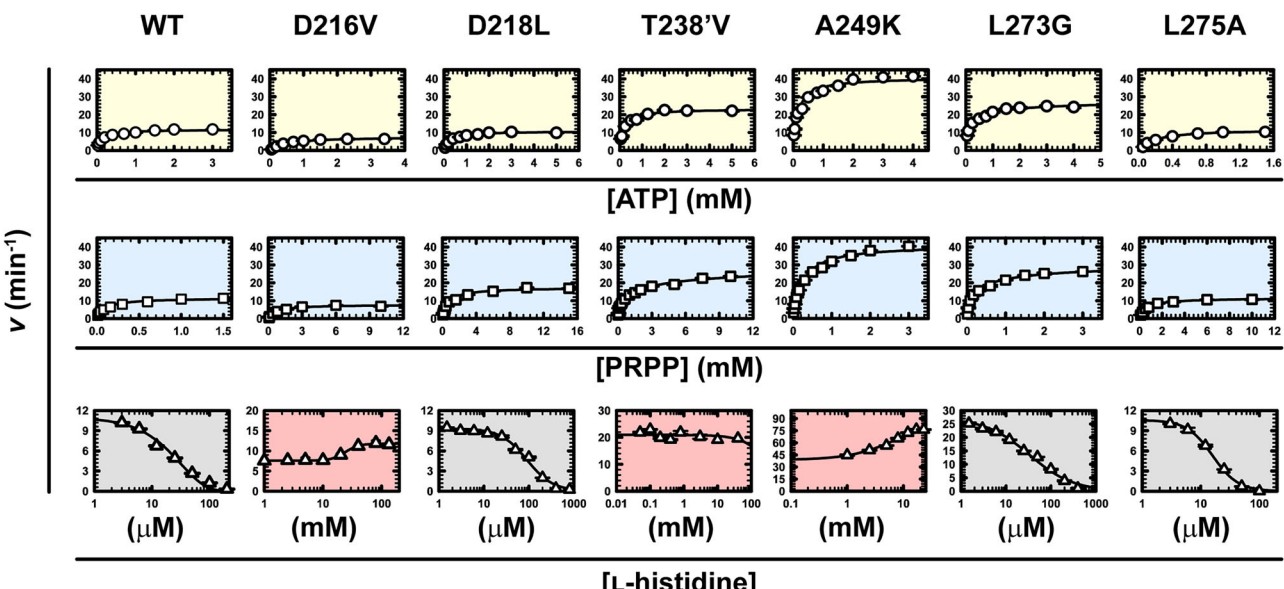

**Fig. 2 | Michaelis-Menten and allosteric kinetics for ATP-PRT variants.** Saturation curves for ATP (light yellow – circles), at saturating concentrations of PRPP, $Mg^{2+}$ and $K^+$. Saturation curves for PRPP (light blue – squares), at saturating concentrations of ATP, $Mg^{2+}$ and $K^+$. Allosteric inhibition (light gray) or activation (light pink) curves for L-histidine (triangles), at saturating concentrations of ATP, PRPP, $Mg^{2+}$ and $K^+$. The concentration range used for L-histidine is shown below each plot, spanning μM and mM ranges, depending on the variant used. The data shown are the average of three replicates representative of two independent experiments. Error bars represent the standard error of the mean.

**Table 1 | Steady-state and allosteric kinetic parameters for wild-type ATP-PRT and variants**

| Variant | $V_{max}$ (s⁻¹) | $K_{m\ ATP}$ (μM) | $K_{m\ PRPP}$ (μM) | $V/K_{ATP}$ (M⁻¹s⁻¹) | $V/K_{PRPP}$ (M⁻¹s⁻¹) | $IC_{50\ L\text{-}His}$ (μM) | $n^H$ | $AC_{50\ L\text{-}His}$ (mM) | $V_F$ (s⁻¹) |
|---|---|---|---|---|---|---|---|---|---|
| WT | 0.16 ± 0.01 | 126 ± 16 | 97 ± 16 | $1.3 \times 10^3$ | $1.6 \times 10^3$ | 21 ± 4 | 1.3 | – | – |
| D216V | 0.13 ± 0.01 | 428 ± 35 | 706 ± 91 | $3.0 \times 10^2$ | $1.8 \times 10^2$ | – | – | 24 ± 1 | 0.2 ± 0.1 |
| D218L | 0.29 ± 0.01 | 269 ± 24 | 709 ± 127 | $1.1 \times 10^3$ | $4.1 \times 10^2$ | 90 ± 8 | 1.5 | – | – |
| T238'V | 0.39 ± 0.01 | 182 ± 22 | 2001 ± 31 | $2.1 \times 10^3$ | $1.9 \times 10^2$ | – | – | – | – |
| A249K | 0.69 ± 0.03 | 153 ± 23 | 251 ± 36 | $4.5 \times 10^3$ | $2.7 \times 10^3$ | – | – | 9 ± 2 | 1.4 ± 0.1 |
| L253'A | – | – | – | – | – | – | – | – | – |
| A273G | 0.53 ± 0.02 | 202 ± 23 | 301 ± 28 | $2.6 \times 10^3$ | $1.7 \times 10^3$ | 33 ± 5 | 0.9 | – | – |
| L275A | 0.20 ± 0.01 | 192 ± 22 | 600 ± 68 | $1.0 \times 10^3$ | $3.3 \times 10^2$ | 16 ± 1 | 2.0 | – | – |
| L275D | – | – | – | – | – | – | – | – | – |

## L-histidine over-production in M. tuberculosis ATP-PRT allosteric variants

To probe the impact of these allosteric variants on the metabolism of *M. tuberculosis* we employed a two-sequential single-crossover strategy[32], introducing point mutations leading to substitutions (D216V, T238'V and A249K, Supplementary Fig. S4), in the *hisG* gene, which encodes ATP-PRT. This approach allows the desired mutations to be introduced while preserving *hisG* at its natural locus, therefore avoiding potential polar effects. The final allosteric variant *hisG* genes were subjected to Sanger sequencing and the mutant strains underwent whole genome sequencing to confirm the introduction only of the desired mutations and lack of changes elsewhere in the genome. As a first control, we evaluated the expression levels of ATP-PRT protein in these variants, using anti-ATP-PRT serum. Importantly, levels of ATP-PRT variants were identical to that of the parent strain as determined by Western blotting (Fig. 3A). This demonstrates that no substantial effects on ATP-PRT protein synthesis, stability or degradation have been caused by the mutations. Therefore, any subsequent results observed must have been triggered by changes in the allosteric kinetics of ATP-PRT.

Next, we evaluated growth kinetics to assess whether allosteric dysregulation of L-histidine biosynthesis reduces fitness and impairs growth. The D216V, T238'V and A249K strains exhibited a reduced growth rate (doubling time) and diminished final biomass after 12 days (Fig. 3B). The reduced growth rate indicates that D216V, T238'V and A249K strains are using nutrients and/or energy faster than the parental strain, thereby starving cells. As the final biomass is also affected in the mutant strains, it is likely that L-histidine, or a metabolite derived from it, accumulates over time, leading to additional toxicity. If this is the case and the reduced growth rate is the dominant factor that limits fitness, these phenotypes should be rescued by growing mutant cells in media with increasing concentrations of nutrients. Indeed, when the most attenuated strain, A249K, was cultured in increasing concentrations of glycerol as a carbon source (Fig. 3C), growth attenuation was decreased, matching the growth of the parent strain. These results are consistent with nutrient depletion constituting the key mechanism of toxicity associated with L-histidine overproduction in *M. tuberculosis*.

Attempts to generate a *hisG* knockout strain in *M. tuberculosis* in the presence of L-histidine supplementation were unsuccessful, suggesting that L-histidine uptake is not sufficient to replace biosynthesis. Of interest, analysis of an *M. smegmatis hisG* knockout strain (Supplementary Fig. S5A) reveals that supplementation does not fully rescue the reduced growth phenotype, confirming that L-histidine uptake in mycobacteria is not sufficient to supply cellular demand. This result

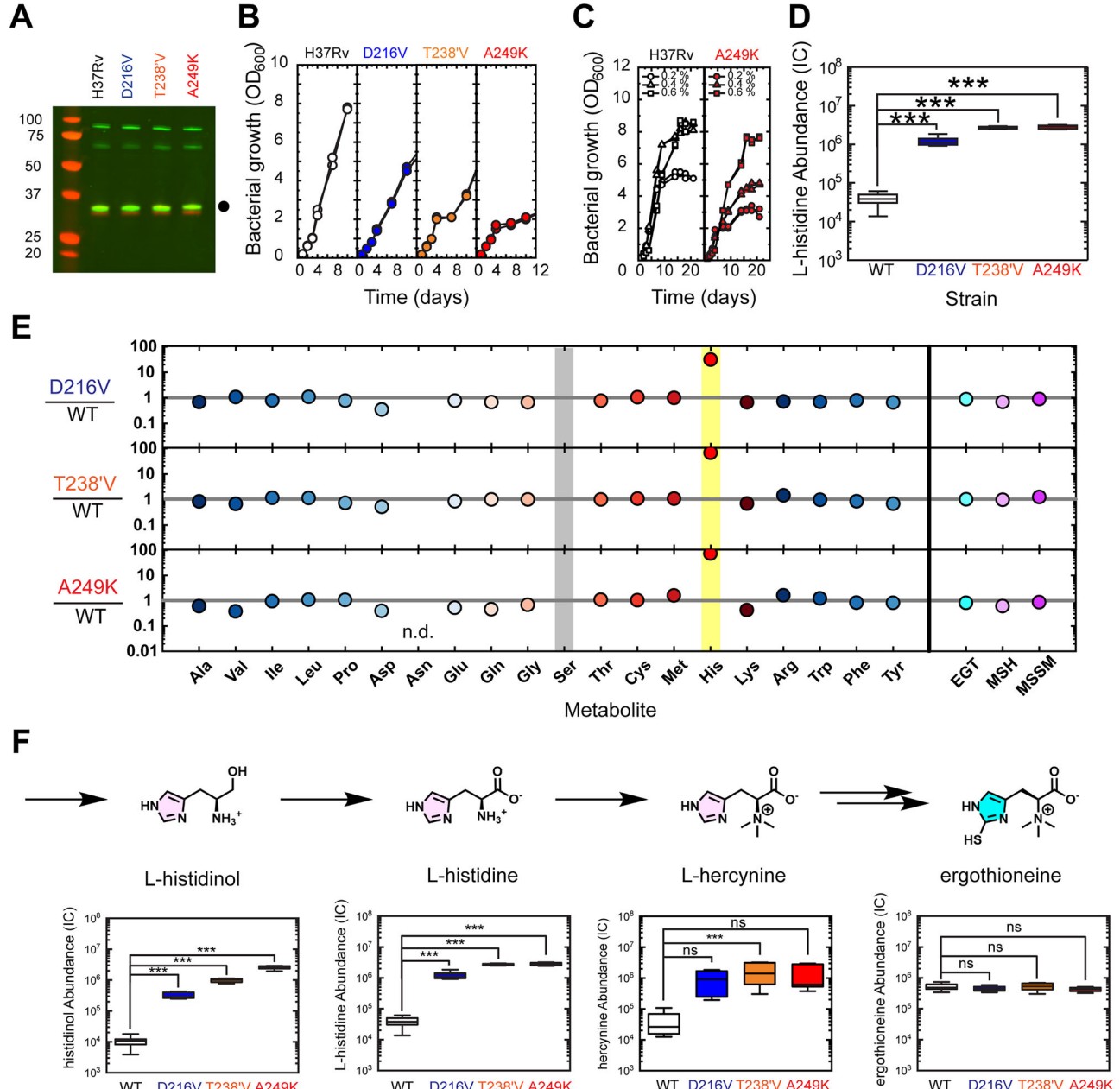

**Fig. 3 | Allosteric variants dysregulate L-histidine production in M. tuberculosis. A** Western blotting with anti-ATP-PRT serum showing a lack of changes in protein levels. Lane 1 molecular weight marker, lane 2 H37Rv, lane 3 D216V, lane 4 T248'V and lane 5 A249K. Anti-antigen 85 serum was used as a loading control. The black dot shows the position of the monomeric ATP-PRT protein (MW = 30.5 kDa). **B** Growth kinetics of wild-type M. tuberculosis H37Rv and the D216V, T248'V and A249K strains in Middlebrook 7H9 complete medium at 37 °C. The data show is from two biological replicates and is representative of three independent experiments. **C** Growth kinetics of H37Rv and the A249K strains in Middlebrook 7H9 medium with increasing amounts of glycerol (0.2, 0.4 and 0.6 %). Final biomass for each strain was lower than for parent strain, however, due to clumping, the data obtained was not plotted. **D** Overproduction of L-histidine in the mutant strains D216V, T248'V and A249K, compared with H37Rv. **E** Relative changes (mutant/wild-

type strain) in the pool size of proteinogenic amino acids and other key metabolites in the D216V, T248'V and A249K strains. No Asn was detected in metabolomics experiments consistent with the knowledge that M. tuberculosis does not synthesize this amino acid in free form. The abundance of Ser is not reported as the peak for this metabolite overlapped with a contaminant. Metabolomic data show is the average ± standard error of three biological replicates and is representative of two independent experiments. **F** Scheme for the biosynthesis of ergothioneine from L-histidine and the pool size of key intermediates in H37Rv and in the D216V, T248'V and A249K strains. Data shown are representative of the mean, whereas error bars represent standard error SEM, n = 3, representative of at least two independent experiments. P-values are indicated as: * P ≤ 0.05, ** P ≤ 0.001, *** P ≤ 0.0001, **** P ≤ 0.00001.

also highlights that L-histidine transport by these two related species is different. Assuming L-histidine export is not possible (supported by the finding that no secretion of L-histidine has been observed), M. tuberculosis growth and fitness are likely to be reduced faster by L-histidine overproduction compared to M. smegmatis, because M. tuberculosis

cannot dispose of L-histidine through degradation or export, as it lacks homologs of genes encoding the required enzymes and transporters[33].

To directly evaluate whether these allosteric variant strains produce more L-histidine than the parent strain, we employed liquid chromatography-mass spectrometry metabolomics[34,35]. Mutations

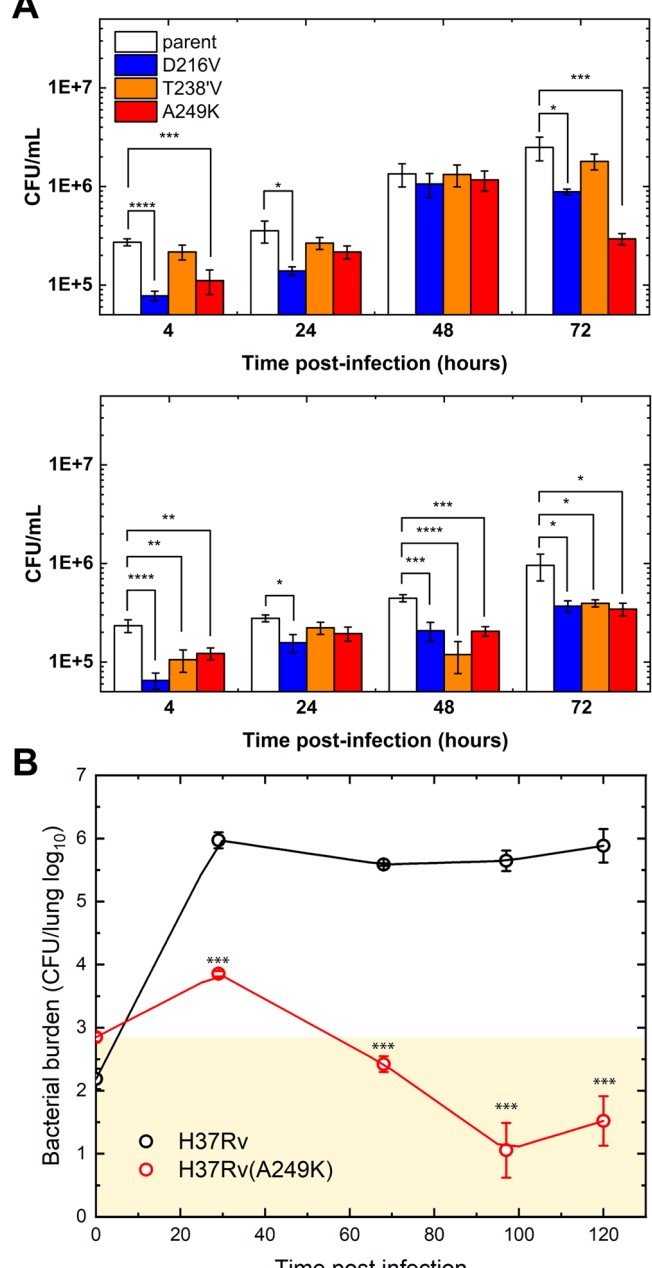

**Fig. 4 | L-histidine overproduction is bactericidal to M. tuberculosis. A** M. tuberculosis uptake and survival inside resting (top panel) or interferon-γ-activated (bottom panel) human blood monocyte derived macrophages of the D216V, T248'V and A249K strains compared with the wild-type H37Rv strain over time. The columns are representative of the mean, whereas error bars represent standard error SEM, n = 3. One-way ANOVA analysis was performed with Dunnett's correction for multiple comparisons. P-values are indicated as: * $P \le 0.05$, ** $P \le 0.001$, *** $P \le 0.0001$, **** $P \le 0.00001$. **B** Low-dose aerosol infection model employing C57/B6 mice. White symbols represent data obtained with the M. tuberculosis H37Rv strain, and red symbols represent data obtained with the A249K strain. Data is from one experiment representative of two independent experiments. Five mice were used per strain per time point. Symbols are the average of CFUs from one lung of five animals, and error bars are the standard error of the mean. An unpaired two-tail t test test was used for comparisons. P-values are indicated as: *** $P \le 0.0001$. Pale yellow box illustrates the area of bacterial killing (below the inoculum).

leading to activated ATP-PRT (D216V, T238'V and A249K) cause a significant increase in concentration of L-histidine by 30- to 70-fold compared to the wild-type strain (Fig. 3D). These results confirm that our rationally designed ATP-PRT allosteric variants result in metabolic dysregulation, bypassing natural feedback inhibition and overproducing L-histidine. To confirm that the increase in L-histidine level is specific, we analyzed the levels of other proteinogenic amino acids. Only modest changes in the levels of other amino acids are observed (Fig. 3E), confirming the specificity of the metabolic disruption imparted by the allosteric variants. In addition, no changes in the levels of EGT, mycothiol (MSH) and mycothione (MSSM) were observed (Fig. 3E), further demonstrating that the phenotypes triggered by these mutations are specific to L-histidine, and not caused by small-molecule thiol imbalance due to changes in EGT levels[36,37]. Curiously, while the levels of EGT are comparable among the strains studied, the levels of hercynine (N,N,N-trimethyl-L-histidine), an intermediate in EGT biosynthesis, mirror the increased amounts of L-histidine (Fig. 3F), pointing to a lack of regulation at the first steps in the pathway. This demonstrates that the regulation of EGT biosynthesis in *M. tuberculosis* lies in the last steps of the pathway and not in the first committed step, as is usually the case. Collectively, these results suggest that L-histidine overproduction-associated loss of fitness, caused by allosteric dysregulation, is largely due to nutrient and/or energy depletion and to a buildup of toxic levels of L-histidine, and not by redox imbalance.

## L-histidine over-production is bactericidal to M. tuberculosis during infection

Motivated by the strong and specific metabolic and microbiological phenotypes observed with these ATP-PRT mutant strains (Fig. 3), we next aimed to demonstrate the overall metabolic cost of allosteric dysregulation of L-histidine biosynthesis during infection. We first employed human blood-derived monocytes, which were differentiated into macrophages. Infection of both resting and interferon-γ-activated human macrophages with our three L-histidine over-producing strains (D216V, T238'V and A249K) led to differences compared to infections with the parent strain (Fig. 4A). Notably, CFUs for all mutants are significantly lower than of the parent strain in the four-hour time, which in our case not only reflects uptake, but also early killing by the macrophages. In unstimulated macrophages, all strains grew to comparable levels for the first 48 h, and then significant reductions in CFU between 0.5 and 1 log10 were observed with T238'V and A249K mutants, compared to the parent strain. In activated macrophages, mutants never reached the levels of the parent strain, but growth persisted at a significantly reduced rate for 72 hours. Levels of L-histidine produced by mutant strains correlate with the attenuation observed (Supplementary Fig. S6), although we only have three mutants to compare.

Next, we selected the most attenuated mutant (A249K) to investigate the impact of L-histidine overproduction in mice using low-dose aerosol infection[38]. Infection of C57BL6 mice with the A249K mutant strain resulted in significant attenuation of *M. tuberculosis* infection compared with the wild-type strain from 30 days after infection (Fig. 4B). By day 70 the bacterial burden rescinds to levels below the inoculum, thereby demonstrating that L-histidine overproduction is bactericidal to *M. tuberculosis* in vivo. CFUs decreased to approximately 1.5 log10/lung (midpoint between CFUs/lung obtained at days 100 and 120 post-infection) from the inoculum, although no sterilization was observed. The multi-log attenuation of infection observed with the A249K strain of *M. tuberculosis* is striking as it parallels or surpasses the effect observed with antibiotics and with deletion of deemed important targets, such as *icl*1, encoding isocitrate lyase[39]. Together, these results demonstrate that allosteric dysregulation of ATP-PRT leads to over-activation of L-histidine biosynthesis, which is mycobactericidal during macrophage and murine infection in mice.

## Discussion

AMR has emerged as one of the most urgent global health challenges, with projections estimating up to 10 million deaths annually by 2050 if new antibiotics are not developed[7]. There is a critical need for novel antibiotics, particularly those effective against resistant bacterial strains such as *M. tuberculosis*, which currently account for approximately over 150,000 deaths annually[8]. It is consensus that novel drugs with novel MoA are urgently needed if we are to revert these trends[1,5,40].

This article presents a new antibiotic discovery strategy based on metabolic activation—leveraging bacterial metabolism as a novel and promising therapeutic avenue. While this has not yet been tested in antibiotic pharmacology, activation of enzymes involved in disease, such as kinases and phosphatases, is an area of growing interest and increasing investigation[41] and several drugs in clinical use are agonizts or activators of proteins[18]. The proof-of-concept target in this study is the L-histidine biosynthesis pathway in *M. tuberculosis*, which is both essential for bacterial growth and highly energy-intensive, requiring around 40 ATP molecules per L-histidine produced. Precisely, we focused on the enzyme ATP phosphoribosyltransferase (ATP-PRT), which catalyzed the first step in the pathway and naturally subject to allosteric inhibition[29,42]. Importantly, we have demonstrated in vitro that ATP-PRT can be activated beyond its natural maximal activity[30,31]. Here, we repurposed the enzyme's allosteric regulatory mechanism—not to inhibit, but to activate its function—effectively reversing its typical regulation and transforming the allosteric "chassis" into a tool for metabolic activation.

Our findings show that allosteric activation of L-histidine bio-synthesis dramatically impairs *Mycobacterium tuberculosis* infection in vivo. Specifically, we demonstrate that overproduction of L-histidine compromises bacterial growth and fitness by driving excessive consumption of nutrients, such as carbon and energy sources. These results suggest that small molecules capable of mimicking this metabolic imbalance could be developed as antibiotics. This strategy represents a novel therapeutic approach based on metabolic agonism —activating, rather than inhibiting, a bacterial metabolic pathway.

The strategy employed in this study—rationally designing and characterizing vulnerabilities associated with the allosteric activation of metabolic enzymes—offers broad applicability for antibiotic discovery and target validation. This approach can be used to identify not only suitable target enzymes and metabolic pathways but also pathogens that are particularly susceptible to metabolic dysregulation. Beyond L-histidine biosynthesis, other pathways, such as those involved in branched-chain or aromatic amino acid biosynthesis, may also be amenable to allosteric manipulation. Notably, resistance to compounds acting via this mechanism would likely come at a high fitness cost. Mutations that block allosteric activation may also impair its natural feedback regulation, potentially destabilizing flux control through the pathway. Supporting this idea, our extensive in vitro and in vivo studies failed to identify any revertant or suppressor mutants capable of overcoming the induced phenotypes. These results need to be formally tested in other enzymes and pathways, and across different species, yet they suggest that targeting the allosteric regulation of key metabolic enzymes could represent a fundamentally unique paradigm for "resistance-proof" antibiotic discovery and development.

## Methods

### Ethical statement

This research was carried out in accordance with all relevant ethical regulations from the Francis Crick Institute and performed under a U.K. Home Office project license PPL P4D8F6075 in accordance with the Animal Scientific Procedures Act, 1986.

### Site-directed mutagenesis

The modified constructs, ATP-PRT[D216V], ATP-PRT[D218L], ATP-PRT[T238'V], ATP-PRT[A249K], ATP-PRT[L253'A], ATP-PRT[A273G], ATP-PRT[L275A] and ATP-PRT[L275D] were obtained by site-directed mutagenesis of the WT expression construct. Primers for mutagenesis were designed for each case following the instructions of the QuickChange method from Agilent Technologies. Sequences of primers used for site-directed mutagenesis are shown below:

D216V_FW − 5′- CAGCAATATCTGATGCTGGTGTACGACTGTCCG CGTTCGG-3′

D216V_RV − 5′- CCGAACGCGGACAGTCGTACACCAGCATCAGATA TTGCTG-3′

D218L_FW − 5′- GCAATATCTGATGCTGGACTACCTGTGTCCGCG TTCGGCTCTG-3′

D218L_RV − 5′ - CAGAGCCGAACGCGGACACAGGTAGTCCAGCAT CAGATATTGC-3′

T238'V_FW − 5′- GGTCTGGAGAGCCCAGTGATTGCCCCGCTGG CGG-3′

T238'V _RV − 5′- CCGCCAGCGGGGCAATCACTGGGCTCTCCA-GACC-3′

A249K_FW − 5′- CCGGACTGGGTGAAAATTCGCGCGCTGGTCC CG-3′

A249K _RV − 5′- CGGGACCAGCGCGCGAATTTTCACCCAGTCC GG-3′

L253'A_FW − 5′- GGCGATTCGCGCGGCAGTCCCGCGTCGTGA TG-3′

L253'A_RV − 5′- CATCACGACGCGGGACTGCCGCGCGAATCGCC-3′

A273G_FW − 5′- GCGATCGGTGCGAAAGGCATCTTGGCGAGCG-3′

A273G_RV − 5′- CGCTCGCCAAGATGCCTTTCGCACCGATCGC-3′

L275A_FW − 5′- GGTGCGAAAGCGATCGCGGCGAGCGATATCCG-3′

L275A _RV − 5′- CGGATATCGCTCGCCGCGATCGCTTTCGCACC-3′

L275D_FW − 5′- GATCGGTGCGAAAGCGATCGATGCGAGCGATAT CCG-3′

L275D _RV − 5′- CGGATATCGCTCGCATCGATCGCTTTCGCACC GATC-3′

Linear amplification of the mutant DNA was achieved using KOD Hot Start DNA polymerase master mix (Merck Millipore), mutagenesis primers at 0.14 μM and 0.2 ng/μl of template DNA. Cycling parameters were 95 °C for 2 min, followed by 20 cycles of 95 °C (20 s); 55 °C (20 s); 70 °C (2.5 min), and finally 72 °C for ten minutes. Template DNA was degraded after amplification using DpnI (New England Biolabs), and the template-free DNA was used to transform library efficiency NEB5α *E. coli* cells (New England Biolabs). Two colonies from each plate were sent for Sanger sequencing to confirm the presence of the mutation.

### ATP-PRT expression and purification

Recombinant wild-type ATP-PRT[WT] and mutants ATP-PRT[D216V], ATP-PRT[D218L], ATP-PRT[T238'V], ATP-PRT[A249K], ATP-PRT[L253'A], ATP-PRT[A273G], ATP-PRT[L275A] and ATP-PRT[L275D] were expressed in BL21(DE3)pLysS using a pJ411 plasmid with an uncleavable 6xHis-tag as described for wild-type ATP-PRT[29]. Frozen cells that expressed the protein were thawed on ice, resuspended in buffer A (20 mM triethanolamine (TEA), pH 7.8, 300 mM NaCl, and 50 mM imidazole containing lysozyme, cOmplete® protease inhibitor cocktail and DNase) and lysed by sonication before centrifugation at 25,000 × g for 30 min. The soluble fraction was loaded into a Ni-NTA column, and the protein separated by a gradient using buffer C [20 mM TEA (pH 7.8), 300 mM NaCl and 500 mM imidazole]. Fractions were analyzed by SDS-PAGE. Fractions containing only ATP-PRT were dialyzed in 20 mM TEA (pH 7.8), concentrated, and stored at − 80 °C after being flash-frozen. Purified ATP-PRT variants were analyzed by SDS-PAGE. This protocol allowed the purification to homogeneity of 5 to 25 mg of each ATP-PRT variant from 5 g of wet cell pellet. The protein concentration was determined by UV absorbance at 280 nm, using calculated extinction coefficients ($\varepsilon_{280} = 20{,}800\ M^{-1}cm^{-1}$).

### CD spectroscopy

CD spectra were recorded on a Jasco J-815 spectro-polarimeter equipped with a Peltier system for temperature control. CD intensities

are presented as the CD absorption coefficient calculated by using the molar concentration of the proteins. Far-UV CD spectra were recorded from 200 to 260 nm, while near-UV spectra were recorded from 260 to 320 nm.

## CD thermal unfolding
Thermal unfolding of ATP-PRT and variants in the near-UV was monitored between 20 and a maximum of 65 °C with a heating rate of 1 °C per minute, and the signal recorded at 270 nm using a concentration of 70 μM of protein. Near-UV ATP-PRT unfolding was monitored in a series of buffers with pHs ranging from 7.0 to 10.5 [in 50 mM MOPS (7.0), HEPES (7.5 and 8.0), TAPS (8.5), CHES (9.0 and 9.5) and CAPS (10.5) buffers] in the presence of 7 mM MgCl2 and 200 mM KCl and in the absence or presence of 0.5 mM L-His. The apparent melting point ($T_m^{app}$) was defined as the temperature where the CD absorption coefficient equals zero. Thermal unfolding of ATP-PRT and ATP-PRT variants in the far-UV was monitored between 20 and 90 °C with a heating rate of 1 °C per minute, and the signal recorded at 225 nm using a concentration of 4.75 μM of protein. Far-UV unfolding of ATP-PRT and ATP-PRT variants were monitored in a buffer containing 50 mM Tris-HCl, pH 8.5, 7 mM MgCl2 and 200 mM KCl and in the presence or absence of 1 mM L-His.

## ATP-PRT kinetics
Steady-state kinetic assays were conducted using a Shimadzu UV-2550 spectrophotometer equipped with dual-beam optics and a Peltier system for temperature control. Initial velocities for the forward reaction of ATP-PRT were measured by following the formation of PR-ATP ($\varepsilon_{290} = 3600$ M-1cm-1), in the presence of inorganic pyrophosphatase or PPiase[29]. A typical reaction mixture contained 50 mM Tris-HCl (pH 8.5), 7 mM MgCl2, 200 mM KCl, 3 mM ATP, 1.5 mM PRPP, 600 nM PPiase and 450 nM ATP-PRT WT or variants. All kinetic assays were conducted at 25 °C ± 0.2 °C.

Steady-state data were fitted using the nonlinear, least-squares, curve-fitting programs of Sigma-Plot for Windows, version 11.0. Individual saturation curves were fit to Eq. 1

$$v = VS/(S + K) \qquad (1)$$

where $V$ is the maximal velocity, $S$ is the substrate concentration, and $K$ is the Michaelis constant for the substrate ($K_m$). Inhibition data obtained under saturating concentrations of substrates and metals, and variable concentration of L-His, were fit to Eq. 2

$$v = v_0/\left[1 + (I/IC_{50})^{n_H}\right] \qquad (2)$$

where $v_0$ is the uninhibited velocity, $I$ is the L-His concentration, $IC_{50}$ is the concentration of L-His necessary to give 50% inhibition and nH is the Hill number.

## Generation of mutations in *Mycobacterium tuberculosis*
A 2832 bp region containing the *hisG* gene was amplified from H37Rv using KAPA HiFi with primers SNP for and SNP rev (Supplementary Table S1). The resulting PCR product was cloned into pCR4-Blunt-TOPO (Invitrogen). This plasmid was used to introduce the mutations into *hisG* using site directed mutagenesis (SDM) Table 1. The resulting plasmids were Sanger sequenced to confirm the mutation and excised with *kpnI* and cloned into p2NIL *kpnI* site. The selection markers for pGOAL17 were introduced into the unique *pacI* site. Electro competent H37Rv cells were transformed with 1 μg of plasmid DNA for each of the mutations and selected on 7H11 plates containing kanamycin (KAN 25 μg/ml), X-gal (X-GAL 50 μg/ml). Blue KAN-resistant colonies were picked onto 7H11 plates containing 2% sucrose and X-GAL. White sucrose tolerant colonies were picked for screening. The colonies were scraped into Instagene mix (BioRad) and screened using MAMA PCR[43]. The correct clones were PCR-amplified across the region using primers *hisG* for and *hisG* rev and Sanger sequenced to confirm the presence of the mutation.

## Bacterial strains and growth condition
All plasmids were cloned and propagated in *E. coli* DH5 alpha in Lysogeny broth (LB) or LB agar containing 25 μg/ml kanamycin (KAN) at 37oC. H37Rv strain was used for homologous recombination experiments. This was grown in Middlebrook 7H9 media containing 10% albumin-dextrose-catalase (ADC), 0.05% tyloxapol and 0.4% glycerol unless stated.

## Growth curves
All manipulations of live *M. tuberculosis* were carried out in Biosafety Level 3 laboratories. Cultures were typically grown at 37 °C in Middlebrook 7H9 medium (Sigma) supplemented with 10% albumin-dextrose-catalase supplement (ADC, Sigma) and 0.05% Tyloxapol (Sigma) or on Middlebrook 7H10 agar medium (Sigma) supplemented with 10% oleic acid-albumin-dextrose-catalase supplement (OADC, Sigma). Mycobacterial growth was carried out in 1 L polycarbonate culture bottles in a Bellco roll-in incubator (2 r.p.m.) at 37 °C containing 100 mL of Middlebrook 7H9 broth with varying amounts of glycerol. Bacterial growth was followed by turbidimetry (OD600).

## Small molecule extracts (SME)
*M. tuberculosis* strains were grown in 7H9 media until OD 1, 100 μL were then transferred onto 0.22 μm nitrocellulose filters and grown on 7H11 agar plates containing ADC for 5 days. The metabolites were extracted by mechanical lysis in cold acetonitrile/methanol/water (2:2:1) containing 0.1 mm acid-washed Zirconia beads. The SME were clarified by centrifugation and filtered through a 0.22 μm Spin-X column (Costar). The SME were mixed 1:1 with acidified acetonitrile (0.1% formic acid). The samples were normalized to protein concentration using the BCA assay kit (Thermo).

## Liquid chromatography - mass spectrometry metabolomics
An Agilent 1290 LC system consisting of a solvent degasser, binary pump, isocratic pump, temperature-controlled auto-sampler and temperature-controlled column compartment equipped with a Cogent Diamond Hydride Type C silica column (150 mm × 2.1 mm; MicroSolv Technology Corporation) was used for liquid chromatography. Solvent A was 0.1% formic acid in water, and solvent B was 0.1% formic acid in acetonitrile. The gradient was as follows: 0–2 min 85% B; 2-3 min to 80% B; 3–5 min 80% B; 5-6 min to 75% B; 6-7 min 75% B; 7-8 min to 70% B; 8-9 min 70% B; 9-10 min to 50% B; 10-11 min 50% B; 11-11.1 min to 20% B; 11.1–14 min hold 20% B. A flow rate of 0.4 mL/min was used. An Agilent Accurate Mass 6230 TOF apparatus was used. Dynamic mass axis calibration was achieved by continuous infusion of a reference mass solution using an isocratic pump connected to a Jet-Stream ionization source, operated in the positive-ion and negative-ion mode. Jet Stream parameters used were: ESI capillary, nozzle and fragmentor voltages were set at 3500 V, 2000 V and 110 V, respectively, drying gas 13 L/min, nebulizer pressure 35 psi, sheath gas temperature 350 °C, and sheath gas flow 12 L/min. The MS acquisition rate was 1.0 spectra/sec, and m/z data ranging from 50-1200 were stored. The instrument routinely enabled accurate mass spectral measurements with an error of less than 5 parts-per-million (ppm), mass resolution ranging from 10,000–25,000 over the m/z range of 121-955 atomic mass units, and a 100,000-fold dynamic range with picomolar sensitivity. Data were collected in the centroid mode in the 4 GHz (extended dynamic range) mode. Statistical significance between means was evaluated with ANOVA using Tukey's range test.

## Cell free extracts (CFE) for Western blot

*M. tuberculosis* strains were grown in 7H9 media until an OD 1 and then inoculated onto 0.22 μm nitrocellulose filters and grown on 7H11 agar plates containing ADC for 5 days. Extracts were prepared by mechanical lysis in PBS buffer containing protein inhibitors (cOmplete® mini EDTA free, Roche) with 0.1 mm acid washed Zirconia beads. The lysates were clarified by centrifugation and filtered through a 0.22 μm Spin-X column (Costar).

## Western blots

10 μg of CFE was run a 4 – 12 % gradient gel and transferred to a nitrocellulose membrane. The membrane was incubated for 1 h in anti-hisG antibody (1/10,000 dilution), washed, and then incubated with goat anti-rabbit secondary antibody (1/10,000 dilution IRDye 800CW, Licor), washed and scanned on an ODYSSEY CLx machine. Antigen 85 (Abcam 36731) was used as a loading control (Ag85 goat anti-mouse IRDye 680RD Licor 92568070).

## Preparation and culture of human monocyte-derived macrophages

Peripheral blood mononuclear cells (PBMCs) from healthy anonymous donors were isolated from leukocytes provided by the National Health Service, Blood and Transplant Service, UK as previously described[44]. Briefly, PBMCs were isolated by centrifugation on Ficoll-Paque Premium gradient (GE Healthcare, 17-5442-03) for 60 min at $300 \times g$ at room temperature. Mononuclear cells were collected and washed twice with MACS rinsing solution (Miltenyi, 130-091-222), and the remaining red cells were treated with 10 mL red blood cell (RBC) lysing buffer (Sigma, R7757) at room temperature for 20 min. Following lysing of red blood cell, monocytes were isolated from PBMCs using a magnetic cell separation system with anti-CD14 magnetic beads (Miltenyi, 130-050-201) on an LS column (Miltenyi, 130-042-401) using a QuadroMACS separator magnet (Miltenyi, 130-090-976). CD14-positive monocytes (106 cells/mL) were cultured in complete RPMI 1640 with GlutaMAX and HEPES (Gibco, 72400-02), 10% fetal bovine serum (Sigma, F7524) containing 10 ng/mL of human granulocyte-macrophage colony-stimulating factor (hGM-CSF) (Miltenyi, 130-093-867) in untreated petri dishes at 37 °C in a humidified 5% CO2 incubator for a total of 6 days, with an equal volume of fresh complete medium, including hGM-CSF was added on day 3. Following differentiation, macrophages were detached using 0.5 mM EDTA in ice-cold PBS and cell scrapers (Sarstedt, 83.1830), pelleted by centrifugation, and resuspended in complete RPMI 1640 medium. Macrophages were then plated in 24-well plates at a concentration of $2 \times 105$ cells/well with complete RPMI 1640 media (for resting macrophages). A subset of macrophages was also treated overnight with recombinant human IFN-γ (PHC 4031; Gibco) at a concentration of 100 μ/mL.

## Macrophage infections with *M. tuberculosis* and *hisG* mutants

*M. tuberculosis* H37Rv and *hisG* mutant strains were grown at 37 °C in Middlebrook 7H9 medium (Difco Laboratories) containing 0.2% glycerol and 0.05% Tween 80 and was supplemented with 10% albumin–dextrose–catalase supplement (ADC; BD) until early logarithmic phase of $OD_{600}$ 0.4–0.8. Single cell suspensions for macrophage infection were prepared by centrifugation of bacterial cultures and sequential washes in PBS and complete RPMI 1640 media. An equal volume of sterile glass beads (2.5–3.5 mm) that matched the pellet size was added, and then the tubes were vigorously shaken for 1 min to break up bacterial aggregates. The bacteria were washed again with complete RPMI 1640 at low centrifugation speed for 5 min. The supernatants containing single cells were transferred into new tubes, the $OD_{600}$ was measured and then diluted to 0.1 in complete RPMI 1640. In our assay, we assumed that an $OD_{600}$ of 1 contains 108 bacteria/mL. Next, macrophages were infected with *M. tuberculosis* strains at an MOI of 1 for 2 h. Following the 2 h uptake, macrophages were

washed thrice with PBS and replaced with fresh complete RPMI 1640 media containing IFN-γ where relevant.

Bacterial viability was determined using CFU counts on agar plates. Briefly, following 2, 24, 48 and 72 h post-infection, macrophages were washed twice with PBS and then lysed in 0.5 mL of water containing Tween-80 (0.05%) for 30 min at room temperature. The lysed solution from triplicate wells was then serially diluted 10-fold into PBS containing Tween-80 (0.05%). 20 μL from each dilution was then plated in triplicates onto Middlebrook 7H11 agar plates containing 0.5% glycerol and supplemented with 10% oleic acid-albumin–dextrose–catalase supplement (OADC; BD). Agar plates were incubated for 3–4 weeks at 37 °C. CFU counts were calculated and plotted as the mean CFU per mL over time. One-way ANOVA analysis was performed with Dunnett's correction for multiple comparisons.

## Murine aerosol *M. tuberculosis* infections

C57BL/6 J mice were bred and maintained under specific pathogen-free conditions at The Francis Crick Institute. All protocols for breeding and experiments were approved by the Francis Crick Institute ethical committee and performed under U.K. Home Office project license PPL P4D8F6075 in accordance with the Animal Scientific Procedures Act, 1986.

Infections were performed in the category 3 animal facilities at the Francis Crick Institute. For aerosol infections, *M. tuberculosis* strains (A249K and parental H37Rv) were grown to mid-log phase in Middlebrook 7H9 broth, containing 10% ADC and 0.05% Tween. An infection sample was prepared from this to enable delivery of approximately 100 CFU/mouse lung using a modified Glas-Col aerosol infection system. Infection was monitored by assessing homogenized lungs from infected mice at defined time intervals. Bacterial counts were determined by plating serial dilutions of homogenates on duplicate Middlebrook 7H11 containing OADC. CFUs were counted 2-3 weeks after incubation at 37 °C. The data at each time point are the means of 5 mice/group +/− SEM. The data is representative of two independent experiments. An unpaired two-tail *t* test test was used for comparisons.

Power calculations: we employed an unpaired *t* test using the 'power calculator' using online software (http://www.biomath.info/power/ttest.htm) to estimate the number of mice required to observe a statistically significant change of 1 log10 colony forming units, with a *p*-value of ≤ 0.05. Based on this change and significance, we would need less than six mice per group to obtain useful data. Therefore, our choice of five mice per group is ideal as it makes the best use of the information and its statistical value obtained with the smallest possible number of animals. Five mice per time point is also the common number of mice employed in several studies with *Mycobacterium tuberculosis*., including several studies we co-authored in established peer-reviewed journals.

## Reporting summary

Further information on research design is available in the Nature Portfolio Reporting Summary linked to this article.

## Data availability

Source data related to Figs. 1d, e, 2, 3b–e, 4a, b are provided as a Source Data file and are also available at Zenodo [https://zenodo.org/records/18232745]. Source data are provided in this paper.

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

## Acknowledgements

We thank Laura Masino (The Francis Crick Institute) for assistance with circular dichroism spectroscopy. Funding: This work was supported by funds from The Francis Crick Institute Core Funding (CC2000 to L.P.S.C.) and (CC2081 to M.G.G.), Wellcome Trust (104785/B/14/Z to L.P.S.C.) and relocation funds provided by the Herbert Wertheim UF Scripps Institute for Biomedical Innovation and Technology (to L.P.S.C.).

## Author contributions

D.M.H., J.P.P., A.R., C.d.C., A.Z., K.L.P., and D.E. performed experiments and analyzed data; A.G.G. performed computational analysis and bioinformatics. S.E., D.S., P.E.B., M.G.G., and L.P.S.C. supervised the work. All authors co-wrote the manuscript. L.P.S.C. conceptualized the work.

## Funding

## Competing interests

The authors declare no competing interests.
