## [Transparent Peer Review file · Nature Communications]

Activation of L-histidine biosynthesis as a new antibiotic strategy against *Mycobacterium tuberculosis*

Corresponding Author: Dr Luiz Pedro de Carvalho

Version 0:

Reviewer comments:

Reviewer #1

(Remarks to the Author)

The manuscript by Hunt et al. describes the design and biochemical characterisation of ATPPRT mutants whose sensitivity to allosteric inhibition by L-histidine is drastically reduced, and in some cases, catalysis is even activated. This builds on their previous groundbreaking discovery of a histidine analogue that, despite binding the same allosteric site and in a similar way as histidine, activates ATPPRT catalysis instead of inhibiting it. Here they also employ *Mtb* genetics and microbiology to introduce to create mutant strains carrying this ATPPRT mutations, and show by MS that they accumulate L-histidine. The mutants have an attenuated growth rate in vitro and in macrophages, and one of them is killed in mice to levels below the initial inoculum, which the authors interpret as indicative of a bactericidal effect of the mutation. This is a fundamental discovery vis-à-vis allosteric regulation of catalysis and bacterium metabolism, and a highly innovative idea to be explored for antibiotic design with a truly novel mode of action. The paper represents a tremendous addition to the field and should of immense interest to the readership of Nat Comm, provided the following questions/comments are addressed.

Abstract

“metabolic activation is a new bactericidal mechanism that could be applied to antibiotic discovery more generally.” I urge more restraint here, unless the authors show evidence that this is the case in other pathogenic bacteria. Otherwise, make the sentence specific to *M. tuberculosis* for now.

Introduction

The statement on page 4 “L-histidine binding to the allosteric domain of *M. tuberculosis* ATP-PRT leads to closing of the hexamer (Fig. 1B) and complete inhibition of the reaction (28-30)” warrants revision. Reading reference 28, nothing is concluded there about closing of the hexamer, and reference 29 argues that it is not the closing of the hexamer that causes inhibition, but a reduction in the frequency of the cycle of loosening and tightening of the hexamer.

Fig 1A (and Fig S1A) should be revisited. The adenine ring of ATP is not ideal, and N1 should not be protonated (change the double bond pattern there). Also, in PRATP, the N6 amino group becomes an imino group at the pH assayed (that's what yields the increase in absorbance at 290 nm; this is correct in Fig 1A). However, in Fig S1A, the adenine C4 and C5 make only 3 bonds each. Most importantly, the structure of PRATP is incorrect as depicted. The easiest way to fix this last point is to invert the stereochemistry depicted for the 3'' and 2'' hydroxyls, and for 5'' carbon.

On p. 3 the paragraph beginning “Human pharmacology includes...” requires references.

Results

The sentence on page 6 “The reduced growth rate indicates that L-histidine overproducers are using nutrients and/or energy faster than parental strain, thereby starving cells” is premature at this point, since it hasn't even been shown at this point that L-histidine is being overproduced, much less that energy misuse is the cause of the reduced growth rate and final biomass. I'd frame it as a hypothesis instead of a forgone conclusion.

Page 6: “Assuming L-histidine export is not possible (supported by the finding that no secretion of L-histidine has been

observed), *M. tuberculosis* growth and fitness are likely to be reduced faster by L-histidine overproduction compared to *M. smegmatis*, because *M. tuberculosis* cannot dispose of L-histidine through degradation (32) or export." Wouldn't this be the case only if L-histidine or a toxic byproduct were the culprits? If lack of nutrients is the main cause of slow growth in the mutants, then getting rid of histidine won't help. Also, is the difference because *M. smegmatis* can export histidine?

On p. 5 "L253'A ... ligand binding does not trigger the conformational change leading to the closed form." L253'A thermal stability profile appears to resemble the closed, more stable, inhibited form of ATPPRT (relative to the wild-type profile) even without histidine present. This is not discussed in the paper. Also, in the paragraph below that, L253'A is bunched with the other mutants whose steady-state kinetics was characterised, but Fig 2 does not show any curves for this mutant, and in Table 1, only dashes are seen instead of kinetic constants values. Was this mutant characterised in terms of steady-state kinetics? Is it active?

The authors suggest *M. tuberculosis* cannot catabolise L-histidine to use it as a source of energy. Is that true? How does *M. tuberculosis* dispose of histidine? One might expect the mutants to be able to use the surplus L-histidine as a source of nutrient, offsetting the detrimental effects seen on the mutants.

Could be massive accumulation of histidine in the mutants alter to pH of the cell? I am not suggesting the authors need to do another experiment, only asking if they know or if they care to speculate.

Minor points:

Bottom of page 3, typo "polypharmacogical"

Be consistent with terms such as PRPP, which appears as "phosphoribosylpyrophosphate" and "phosphoribosylpyrophosphate"

There is no reason to hyphenate ATP-PRT. It should be ATPPRT.

In the SI, "inorganic pyrophosphate" should be "inorganic pyrophosphatase".

On page 5, L235'A should be L253'A

Some figures in the main text require the individual panels of the figure to be labelled (a,b,c...).

Fig 2, if data points are based on averages, then error bars should be shown. Unless they are shown but are just too small?

Table 1, AC50 values, the values and errors should be rounded to the nearest integers.

Reviewer #2

(Remarks to the Author)

In this manuscript, Hunt et al. report a novel approach to combat the *Mtb* pathogen by harnessing allosteric activation of the ATP-phosphoribosyltransferase (ATP-PRT), an enzyme responsible for the initial step of L-histidine biosynthesis. The authors identified amino acid residues crucial for the allosteric regulation of ATP-PRT, constructed, purified, and biochemically characterized several ATP-PRT allosteric mutants with impaired feedback inhibition regulation. Notably, the ATP-PRT D216V, T238V, and A249K variants with mutations in the L-histidine binding site exhibited remarkable resistance to L-histidine inhibition. This simulated allosteric dysregulation, led to a profound increase in L-histidine production in *Mtb*, which impaired its growth. Additionally, the authors showed the bactericidal effect of allosteric activation of ATP-PRT during infection using a macrophage model and a murine infection model.

The approach described in the manuscript is novel in a way that it reports a potential allosteric activation of a target enzyme rather than its inhibition to kill *Mtb*. This antibiotic strategy can thus advance the antituberculosis drug development as the antitubercular drugs acting as allosteric modulators are not yet clinically used in the TB therapy. Apart of that, this study also brings new knowledge about the mechanism of allosteric regulation of mycobacterial ATP-PRT.

Specific comments

1. The authors report that overproduction of L-histidine hinders the growth and fitness of bacteria due to uncontrolled use of nutrients, as demonstrated by glycerol supplementation using the A249K mutant strain (Fig. 3). They conclude: "These results are consistent with nutrient depletion constituting the key mechanism of toxicity associated with L-histidine overproduction in *M. tuberculosis*" and this conclusion is presented also at other places in the manuscript. Have the authors conducted any other analyses to directly show that the mutations in ATP-PRT affect the central carbon metabolism in *Mtb* (e.g. evaluation of ATP levels or metabolomics experiments – beyond the assessment of the amino acid content)? In fact, they report only modest changes in the levels of other proteinogenic amino acids in the mutant strains (although perhaps presentation of the data on log scale might, in part, mask the extent of these changes) (fig. 3). One would expect that global depletion of energy or carbon supplies would affect also production of other amino acids, but the authors claim that this is not the case. Another option for the toxicity of L-His overproduction could be interference specifically with the metabolism of nucleic acids or cell wall due to PRPP depletion, as this molecule (and an ATP-PRT substrate) is the precursor for the synthesis of both. We believe that e.g. evaluation of the cell wall integrity would be informative in this regard.

2. We propose to present the growth kinetics data (Fig. 3B) up to the stationary phase of each strain, which would resolve if the strains just grew more slowly, or if the final biomass (at stationary phase) is affected, as well (in fact, it appears that it is, at least for the strain A249K in 0.2 % glycerol, or 0.4% glycerol, shown in Fig. 3c).

3. We miss the “glycerol rescue” data for the strains D216V and T238’ (Fig. 3C). They should be included to get the complete picture of the described phenomenon and its relevance, especially since the result of this experiment is one of the main arguments to support the nutrient depletion as the mechanism of L-his toxicity.

4. What is the explanation for the significantly lower uptake of the mutant forms of Mtb by both resting and activated macrophages? Specifically, at 4h time-point post-infection one would not expect such differences.

5. The discussion is quite brief. In fact, there is not a single reference included in this part of the manuscript, which is rather unusual. We believe that providing a broader context for the results with already published data would improve the manuscript. For example, are there any known allosteric modulators for the bacterial ATP-PRT enzymes (apart from the one mentioned in ref. 29)? In addition, some of the comments mentioned above could be addressed here.

Technical comments:

1. Fig. 3 B,C,E - the figure legend lacks information whether the results represent the mean of several replicates or whether it is a representative result from several (how many?) independent experiments. Please, add this information.

2. The labelling of the panels (A, B, C, etc.) in the main figures (Fig. 1, Fig. 3 and Fig. 4) is missing. Please, add them in the figures.

3. A few typos:

a. Pg. 4 in the main text – “A216” instead of D216

b. Part Macrophage infections... in Supplementary materials – “singel”, “singe”, “108”

4. We were not able to find the following information in the referred article: “M. tuberculosis cannot dispose of L-histidine through degradation (32) or export.”

Reviewer #3

(Remarks to the Author)

Reviewer #4

(Remarks to the Author)

The observation that allosteric activation of L-histidine biosynthesis is detrimental to M. tuberculosis (Mtb) infection is interesting but I do not think that it is particularly informative in regards to selecting new targets for drug design. This method for drug targeting strikes me as unlikely to be effective. Besides the usual ways that drug resistance develops (target mutation, efflux, degradation, etc) there are likely many additional ways for Mtb to overcome allosteric activation and return to normal levels of L-histidine biosynthesis, for example by limiting other parts of the biosynthesis pathway, so the barrier to resistance is likely to be very low. Also, there are no drugs of this type of the horizon. Additionally, there are numerous examples of Mtb genes that are toxic to the cell when overexpressed. So toxic in fact, that the overexpression plasmid is not even stably expressed in the cell, or if expressed becomes rapidly mutated. So the concept that activating enzymatic pathways can be deleterious to the Mtb cell is far from unique

Reviewer #5

(Remarks to the Author)

This study investigates a novel approach to antibiotic discovery by targeting L-histidine biosynthesis in MTB. The authors engineered allosteric mutants of ATP-phosphoribosyltransferase (ATP-PRT), crucial in L-histidine biosynthesis, to overcome natural feedback inhibition and trigger excessive L-histidine production. This upregulation induced metabolic stress, reducing bacterial growth and fitness. Key findings demonstrated that MTB strains overproducing L-histidine showed diminished infectivity in human macrophages and a mouse infection model, positioning metabolic activation as a promising bactericidal mechanism.

Points of discussion

1. Human Macrophage Model Analysis: the study's infection assays in human blood-derived monocytes highlighted significant differences between the allosteric mutants and the wild-type strain. Mutants, especially the A249K variant,

exhibited reduced uptake and growth in macrophages. Results suggest that nutrient depletion and energy imbalance driven by L-histidine overproduction hinder MTB survival within host cells. However, the lower initial uptake raises questions about whether these mutants are inherently less invasive or if macrophages possess some recognition mechanism affecting mutant strains' entry. It is also surprising that activated Macs don't take up more (which is what they usually do). Also, the display is odd (why the bars and not simple dot-plots? They don't add any information here).

2. Mouse Infection Model Analysis: the mouse model findings revealed a substantial reduction in bacterial load for the A249K variant, with CFU levels dropping below the inoculum level by day 70. One critical aspect omitted is whether compensatory mechanisms in the host (e.g., immune response modifications) influence these outcomes. Further analysis could include macrophage depletion studies or histopathology to ascertain immune involvement in bacterial clearance.

3. Host-Pathogen Interaction: the host section underscores nutrient depletion as a primary toxicity mechanism; however, it does not investigate into potential toxic intermediates or the host's response to L-histidine overproduction. Complementary studies exploring metabolic intermediates or cellular stress responses in macrophages would clarify the broader impact on host-pathogen dynamics. Likewise, I really would like to see the actual CFU data and not only relative changes in the mouse model. Overall, it seems like as if the modified strain is barely able to establish an infection at all. If the absolute amount at day zero is 80 CFU and on day 20 something is 160, the relevance of these findings are questionable.

4. Data Representation and Clarity: the CFU data presentation could benefit from a clearer breakdown of statistical variations between different time points, especially in macrophage studies. A dose-response relationship between the amount of L-histidine produced and the reduction in bacterial load would also improve interpretation. Same is true for mouse infection studies, see comment above.

5. No host component in the mouse model: the host component lacks an in-depth discussion of how L-histidine synthesis pathways might cross-regulate other essential bacterial pathways during host infection. Such exploration would expand on the observed fitness deficits, offering a more comprehensive perspective on bacterial metabolism and potential synergistic drug targets.

REVIEWER COMMENTS (NCOMMS-24-60044-T) and our answers in blue.

Reviewer #1 (Remarks to the Author):

The manuscript by Hunt et al. describes the design and biochemical characterisation of ATPPRT mutants whose sensitivity to allosteric inhibition by L-histidine is drastically reduced, and in some cases, catalysis is even activated. This builds on their previous groundbreaking discovery of a histidine analogue that, despite binding the same allosteric site and in a similar way as histidine, activates ATPPRT catalysis instead of inhibiting it. Here they also employ Mtb genetics and microbiology to introduce to create mutant strains carrying this ATPPRT mutations, and show by MS that they accumulate L-histidine. The mutants have an attenuated growth rate in vitro and in macrophages, and one of them is killed in mice to levels below the initial inoculum, which the authors interpret as indicative of a bactericidal effect of the mutation. This is a fundamental discovery vis-à-vis allosteric regulation of catalysis and bacterium metabolism, and a highly innovative idea to be explored for antibiotic design with a truly novel mode of action. The paper represents a tremendous addition to the field and should of immense interest to the readership of Nat Comm, provided the following questions/comments are addressed.

Thank for the encouraging words and for the critical review of the manuscript and work.

Abstract

“metabolic activation is a new bactericidal mechanism that could be applied to antibiotic discovery more generally.” I urge more restraint here, unless the authors show evidence that this is the case in other pathogenic bacteria. Otherwise, make the sentence specific to M. tuberculosis for now.

Agreed and this sentence now reads “Thus, metabolic activation represents a new mycobactericidal mechanism that could be applied to antimycobacterial drug discovery”.

Introduction

The statement on page 4 “L-histidine binding to the allosteric domain of M. tuberculosis ATP-PRT leads to closing of the hexamer (Fig. 1B) and complete inhibition of the reaction (28-30)” warrants revision. Reading reference 28, nothing is concluded there about closing of the hexamer, and reference 29 argues that it is not the closing of the

hexamer that causes inhibition, but a reduction in the frequency of the cycle of loosening and tightening of the hexamer.

Agreed. Corrected as per reviewers' suggestion. It now reads "L-histidine binding to the allosteric domain of *M. tuberculosis* ATP-PRT leads to stabilization of the closed form of the hexamer, due to reduction of the frequency of open to close transition (Fig. 1B) and complete inhibition of the reaction (28-30).".

Fig 1A (and Fig S1A) should be revisited. The adenine ring of ATP is not ideal, and N1 should not be protonated (change the double bond pattern there). Also, in PRATP, the N6 amino group becomes an imino group at the pH assayed (that's what yields the increase in absorbance at 290 nm; this is correct in Fig 1A). However, in Fig S1A, the adenine C4 and C5 make only 3 bonds each. Most importantly, the structure of PRATP is incorrect as depicted. The easiest way to fix this last point is to invert the stereochemistry depicted for the 3' and 2' hydroxyls, and for 5' carbon.

Apologies for this mistake. A previous, incorrect version of the figure was used. This figure has been swapped by the correct one.

On p. 3 the paragraph beginning "Human pharmacology includes..." requires references.

Referenced now.

Results

The sentence on page 6 "The reduced growth rate indicates that L-histidine overproducers are using nutrients and/or energy faster than parental strain, thereby starving cells" is premature at this point, since it hasn't even been shown at this point that L-histidine is being overproduced, much less that energy misuse is the cause of the reduced growth rate and final biomass. I'd frame it as a hypothesis instead of a forgone conclusion.

Agreed. Fixed. Thanks for pointing this out.

Page 6: "Assuming L-histidine export is not possible (supported by the finding that no secretion of L-histidine has been observed), *M. tuberculosis* growth and fitness are likely to be reduced faster by L-histidine overproduction compared to *M. smegmatis*, because *M. tuberculosis* cannot dispose of L-histidine through degradation (32) or

export." Wouldn't this be the case only if L-histidine or a toxic byproduct were the culprits? If lack of nutrients is the main cause of slow growth in the mutants, then getting rid of histidine won't help. Also, is the difference because *M. smegmatis* can export histidine?

Yes, correct. If the phenotypes observed were solely due to nutrient depletion, differences in L-histidine export would not be relevant.

On p. 5 "L253'A ... ligand binding does not trigger the conformational change leading to the closed form." L253'A thermal stability profile appears to resemble the closed, more stable, inhibited form of ATPPRT (relative to the wild-type profile) even without histidine present. This is not discussed in the paper. Also, in the paragraph below that, L253'A is bunched with the other mutants whose steady-state kinetics was characterised, but Fig 2 does not show any curves for this mutant, and in Table 1, only dashes are seen instead of kinetic constants values. Was this mutant characterised in terms of steady-state kinetics? Is it active?

The L253 mutant was inactive. We could not assay it in any way. And the reviewer is right, it does seem like the L253' mutant is in the closed conformation. We added text clarifying that. It reads as follows: "... , yet it appears to be already closed. Consistent with this interpretation, this mutant was inactive and not investigated further".

The authors suggest *M. tuberculosis* cannot catabolise L-histidine to use it as a source of energy. Is that true? How does *M. tuberculosis* dispose of histidine? One might expect the mutants to be able to use the surplus L-histidine as a source of nutrient, offsetting the detrimental effects seen on the mutants.

Yes, that is true, *M. tuberculosis* is not equipped with the enzymes necessary to break down L-histidine, such as histidine ammonia-lyase and urocanate hydratase and so on. Therefore, it is not surprising that it cannot degrade it or utilize it as a nutrient.

Could be massive accumulation of histidine in the mutants alter to pH of the cell? I am not suggesting the authors need to do another experiment, only asking if they know or if they care to speculate.

Considering that the pK_a of free histidine is close to neutrality (approximately 6.0), and that there will be a significant amount of other free amino acids and proteins, it is unlikely that the intrabacterial pH will be drastically disturbed by histidine overproduction. If it was aspartate, glutamate, lysine or arginine, that could be possible, depending on the amount in question. In agreement with this view, addition of glycerol

rescues the toxicity of the mutants, indicating that the increased L-histidine is not necessarily toxic.

Minor points:

Bottom of page 3, typo "polypharmacogical"

Fixed.

Be consistent with terms such as PRPP, which appears as "phosphoribosylpyrophosphate" and "phosphoribosyl-pyrophosphate"

Thank you, fixed.

There is no reason to hyphenate ATP-PRT. It should be ATPPRT.

Indeed, both forms ATP-PRT and ATPPRT have been used in the literature. We opted to use the hyphenated form, to separate the substrate (ATP) from the reaction (PRT). We decided to keep this form for consistency, as we are others have used in prior publications.

In the SI, "inorganic pyrophosphate" should be "inorganic pyrophosphatase".

Fixed.

On page 5, L235'A should be L253'A

Fixed.

Some figures in the main text require the individual panels of the figure to be labelled (a,b,c...).

Thanks.

Fig 2, if data points are based on averages, then error bars should be shown. Unless they are shown but are just too small?

Indeed, all errors are shown, just too small to be easily seen. We added information about that in the figure legend.

Table 1, AC50 values, the values and errors should be rounded to the nearest integers.

Done. Thanks for catching that.

Reviewer #2 (Remarks to the Author):

In this manuscript, Hunt et al. report a novel approach to combat the Mtb pathogen by harnessing allosteric activation of the ATP-phosphoribosyltransferase (ATP-PRT), an enzyme responsible for the initial step of L-histidine biosynthesis. The authors identified amino acid residues crucial for the allosteric regulation of ATP-PRT, constructed, purified, and biochemically characterized several ATP-PRT allosteric mutants with impaired feedback inhibition regulation. Notably, the ATP-PRT D216V, T238V, and A249K variants with mutations in the L-histidine binding site exhibited remarkable resistance to L-histidine inhibition. This simulated allosteric dysregulation, led to a profound increase in L-histidine production in Mtb, which impaired its growth. Additionally, the authors showed the bactericidal effect of allosteric activation of ATP-PRT during infection using a macrophage model and a murine infection model. The approach described in the manuscript is novel in a way that it reports a potential allosteric activation of a target enzyme rather than its inhibition to kill Mtb. This antibiotic strategy can thus advance the antituberculosis drug development as the antitubercular drugs acting as allosteric modulators are not yet clinically used in the TB therapy. Apart of that, this study also brings new knowledge about the mechanism of allosteric regulation of mycobacterial ATP-PRT.

We thank the reviewers for this critical yet positive appraisal and for highlighting the novelty of the work.

Specific comments

1. The authors report that overproduction of L-histidine hinders the growth and fitness of bacteria due to uncontrolled use of nutrients, as demonstrated by glycerol supplementation using the A249K mutant strain (Fig. 3). They conclude: "These results are consistent with nutrient depletion constituting the key mechanism of toxicity associated with L-histidine overproduction in M. tuberculosis" and this conclusion is presented also at other places in the manuscript. Have the authors conducted any other analyses to directly show that the mutations in ATP-PRT affect the central carbon metabolism in Mtb (e.g. evaluation of ATP levels or metabolomics experiments – beyond the assessment of the amino acid content)? In fact, they report only modest changes in the levels of other proteinogenic amino acids in the mutant strains (although perhaps presentation of the data on log scale might, in part, mask the extent of these changes) (fig. 3). One would expect that global depletion of energy or carbon supplies would

affect also production of other amino acids, but the authors claim that this is not the case. Another option for the toxicity of L-His overproduction could be interference specifically with the metabolism of nucleic acids or cell wall due to PRPP depletion, as this molecule (and an ATP-PRT substrate) is the precursor for the synthesis of both. We believe that e.g. evaluation of the cell wall integrity would be informative in this regard.

The concentrations of various key metabolites, such as amino acids and nucleotides are homeostatically controlled, via mechanisms involving, for example, transcriptional and feedback regulation. That is, their concentration would not fall below certain threshold, as protein synthesis (accounting for most demand) would also stop. Therefore, it is unlikely that we would observe changes in the concentration of these, upon overexpressing one pathway. Considering the unambiguous result obtained with increased glycerol supplementation, i.e., more carbon less grow inhibition, we did not perform additional experiments to confirm this observation.

2. We propose to present the growth kinetics data (Fig. 3B) up to the stationary phase of each strain, which would resolve if the strains just grew more slowly, or if the final biomass (at stationary phase) is affected, as well (in fact, it appears that it is, at least for the strain A249K in 0.2 % glycerol, or 0.4% glycerol, shown in Fig. 3c).

The reviewer is of course correct and that is what we usually do. Unfortunately, we observed significant clumping once these cultures reached stationary phase. But to be clear, the final biomass was significantly reduced and therefore we observed slowdown of growth and strains did not reach the same biomass. We have clarified that on legend of the figure.

3. We miss the "glycerol rescue" data for the strains D216V and T238' (Fig. 3C). They should be included to get the complete picture of the described phenomenon and its relevance, especially since the result of this experiment is one of the main arguments to support the nutrient depletion as the mechanism of L-his toxicity.

We did not perform this experiment with the other strains, as these were less affected than A249K. These experiments were performed in roller bottle incubators and unfortunately, due to numbers of strains and conditions we could not carry them out in parallel. Considering that these are point mutations on the same enzyme, as opposed to deletion of different genes, we surmised that showing rescue for the most attenuated strain would be sufficient. If these were polymorphisms or deletions in different genes, then we would have to have done the experiment in multiple mutants.

4. What is the explanation for the significantly lower uptake of the mutant forms of Mtb

by both resting and activated macrophages? Specifically, at 4h time-point post-infection one would not expect such differences.

Thank you for raising this point. As we also noted in response to another reviewer, the design of our experiment may have led to some confusion. Specifically, bacteria were added to macrophages and incubated for two hours, followed by washing to remove extracellular bacteria, and then incubated for an additional two hours before plating for the four-hour time point. As a result, this time point reflects not only bacterial uptake but also early killing by macrophages. Given this, it is not unexpected that mutants with reduced fitness compared to wild-type show a modest decrease in CFUs under these conditions.

To clarify this in the manuscript, we have revised the text to read:

“Notably, CFUs for all mutants are significantly lower than those of the parent strain at the four-hour time point, which in our case reflects not only uptake but also early killing by macrophages.”

5. The discussion is quite brief. In fact, there is not a single reference included in this part of the manuscript, which is rather unusual. We believe that providing a broader context for the results with already published data would improve the manuscript. For example, are there any known allosteric modulators for the bacterial ATP-PRT enzymes (apart from the one mentioned in ref. 29)? In addition, some of the comments mentioned above could be addressed here.

Agreed and we have expanded the discussion substantially to improve our manuscript. Copied below.

AMR has emerged as one of the most urgent global health challenges, with projections estimating up to 10 million deaths annually by 2050 if new antibiotics are not developed (7). There is a critical need for novel antibiotics, particularly those effective against resistant bacterial strains such as *M. tuberculosis*, which currently account for approximately over 150,000 deaths annually (8). It is consensus that novel drugs with novel MoA are urgently needed if we are to revert these trends (1, 5, 40).

This article presents a new antibiotic discovery strategy based on metabolic activation—leveraging bacterial metabolism as a novel and promising therapeutic avenue. While this has not yet been tested in antibiotic pharmacology, activation of enzymes involved in disease, such as kinases and phosphatases, is an area of growing interest and increasing investigation (41) and several drugs in clinical use are agonists or activators of proteins (18). The proof-of-concept target in this study is the L-histidine biosynthesis pathway in *M. tuberculosis*, which is both essential for bacterial growth and highly energy-intensive, requiring around 40 ATP molecules per L-histidine produced.

Precisely, we focused on the enzyme ATP phosphoribosyltransferase (ATP-PRT), which catalyzed the first step in the pathway and naturally subject to allosteric inhibition (29, 42). Importantly, we have demonstrated in vitro that ATP-PRT can be activated beyond its natural maximal activity (30, 31). Here, we repurposed the enzyme's allosteric regulatory mechanism—not to inhibit, but to activate its function—effectively reversing its typical regulation and transforming the allosteric “chassis” into a tool for metabolic activation.

Our findings show that allosteric activation of L-histidine biosynthesis dramatically impairs *Mycobacterium tuberculosis* infection in vivo. Specifically, we demonstrate that overproduction of L-histidine compromises bacterial growth and fitness by driving excessive consumption of nutrients, such as carbon and energy sources. These results suggest that small molecules capable of mimicking this metabolic imbalance could be developed as antibiotics. This strategy represents a novel therapeutic approach based on metabolic agonism—activating, rather than inhibiting, a bacterial metabolic pathway.

The strategy employed in this study—rationally designing and characterizing vulnerabilities associated with the allosteric activation of metabolic enzymes—offers broad applicability for antibiotic discovery and target validation. This approach can be used to identify not only suitable target enzymes and metabolic pathways but also pathogens that are particularly susceptible to metabolic dysregulation. Beyond L-histidine biosynthesis, other pathways such as those involved in branched-chain or aromatic amino acid biosynthesis may also be amenable to allosteric manipulation. Notably, resistance to compounds acting via this mechanism would likely come at a high fitness cost. Mutations that block allosteric activation may also impair its natural feedback regulation, potentially destabilizing flux control through the pathway. Supporting this idea, our extensive in vitro and in vivo studies failed to identify any revertant or suppressor mutants capable of overcoming the induced phenotypes. These results need to be formally tested in other enzymes and pathways, and across different species, yet they suggest that targeting the allosteric regulation of key metabolic enzymes could represent a fundamentally new paradigm for “resistance-proof” antibiotic discovery and development.

Technical comments:

1. Fig. 3 B,C,E - the figure legend lacks information whether the results represent the mean of several replicates or whether it is a representative result from several (how many?) independent experiments. Please, add this information.

Added, apologies for this omission.

2. The labelling of the panels (A, B, C, etc.) in the main figures (Fig. 1, Fig. 3 and Fig. 4) is missing. Please, add them in the figures.

Fixed.

3. A few typos:

Fixed, thanks.

a. Pg. 4 in the main text – “A216” instead of D216

b. Part Macrophage infections... in Supplementary materials – “singel”, “singe”, “108

4. We were not able to find the following information in the referred article: “M. tuberculosis cannot dispose of L-histidine through degradation (32) or export.”

Apologies for the lack of clarity on this part. This reference merely indicates that the genes encoding enzymes involved in histidine degradation and known transporters are not present in the *M. tuberculosis* genome. We have now expanded for clarity, it now reads “... *M. tuberculosis* cannot dispose of L-histidine through degradation or export, as it lacks homologs of genes encoding the required enzymes and transporters (32).”.

Reviewer #3 (Remarks to the Author):

Comments missing other than what is above.

Reviewer #4 (Remarks to the Author):

We thank the reviewer for the evaluation of our manuscript.

The observation that allosteric activation of L-histidine biosynthesis is detrimental to *M. tuberculosis* (Mtb) infection is interesting but I do not think that it is particularly informative in regards to selecting new targets for drug design.

We agree that activation of a metabolic pathway—specifically histidine biosynthesis in this case—is a compelling concept. However, its relevance to drug design can only be

fully assessed once a small molecule capable of recapitulating this activation is identified. What our study demonstrates is that activation of a metabolic pathway can be as detrimental to bacterial viability as inhibition, and in some cases, even more so. This is both impactful and novel, and to our knowledge, no similar reports exist in other bacterial pathogens. By highlighting this underexplored mechanism, we aim to broaden the scope of antibiotic discovery beyond traditional inhibitory strategies. This method for drug targeting strikes me as unlikely to be effective. Besides the usual ways that drug resistance develops (target mutation, efflux, degradation, etc) there are likely many additional ways for Mtb to overcome allosteric activation and return to normal levels of L-histidine biosynthesis, for example by limiting other parts of the biosynthesis pathway, so the barrier to resistance is likely to be very low.

We appreciate the reviewer's concerns and would like to clarify that our three mutants did not exhibit resistance in the conventional sense. Notably, we have never isolated revertants, which would be a straightforward way for the bacteria to escape the fitness cost associated with these mutations. Based on this, we remain unconvinced that our approach is inherently "unlikely to be effective." In fact, we believe that allosteric activation is as viable a strategy as allosteric inhibition.

The reviewer's concerns regarding potential resistance mechanisms—such as target mutation, efflux, or degradation—are valid and well-recognized in antibiotic development. However, these same mechanisms apply equally to inhibitors of biochemical pathways, many of which have proven to be successful antibiotics. Our study introduces a conceptually distinct approach, and while challenges remain, we believe it merits exploration alongside traditional strategies.

Also, there are no drugs of this type of the horizon.

Given the current landscape—marked by a critical shortage of novel antibiotics and a stagnation in discovery pipelines dominated by analogs of existing drugs—we believe that innovation is essential. The absence of therapeutics that act by activating metabolic pathways underscores the need to explore fundamentally different strategies. Rather than contributing yet another β -lactam derivative, which may be rendered ineffective by existing β -lactamases, our approach seeks to identify new targets, leverage novel chemistry, and explore alternative therapeutic modalities. As discussed in the introduction, allosteric activation and agonism are well-established concepts in pharmacology, though they have not been widely applied in antibiotic development. Our study demonstrates that activating specific enzymes and pathways can be a viable and innovative direction for antibiotic discovery.

Additionally, there are numerous examples of Mtb genes that are toxic to the cell when overexpressed.

We appreciate the reviewer's comment and would like to clarify a key point that may have been misunderstood. Our study does not involve gene overexpression. Rather, we focus on modulating the activity of an enzyme at its native, physiological level. This is fundamentally different from increasing protein abundance through overexpression. In fact, selectively altering gene expression with a small molecule is technically challenging and rarely feasible, especially when protein levels are tightly regulated.

To clarify this distinction in the manuscript, we have added the following text:

“Importantly, activation and agonism are distinct from overexpression of a target, as they work with native, physiological levels of the target protein. While activation and agonism can be straightforwardly generated by a small molecule, specific overexpression of a gene by a small molecule is different and only possible when the levels of the protein in question are not tightly regulated.”

So toxic in fact, that the overexpression plasmid is not even stably expressed in the cell, or if expressed becomes rapidly mutated. So the concept that activating enzymatic pathways can be deleterious to the Mtb cell is far from unique

We appreciate the reviewer's comment and would like to clarify a key distinction. Activating the catalytic rate of an enzyme at physiological levels is fundamentally different from overexpressing a gene. Overexpression alters protein abundance, whereas our approach modulates enzymatic activity without changing protein concentration. This distinction is central to our study, and we carefully controlled for protein levels, as shown in Figure 3A.

We continue to view this work as the first description of metabolic pathway activation in *M. tuberculosis*—and, to our knowledge, in any pathogenic bacterium—leading to a loss of viability during infection. Importantly, the observed effects are not due to protein overexpression, but rather to altered enzymatic activity within physiological bounds.

Reviewer #5 (Remarks to the Author):

This study investigates a novel approach to antibiotic discovery by targeting L-histidine biosynthesis in MTB. The authors engineered allosteric mutants of ATP-phosphoribosyltransferase (ATP-PRT), crucial in L-histidine biosynthesis, to overcome natural feedback inhibition and trigger excessive L-histidine production. This upregulation induced metabolic stress, reducing bacterial growth and fitness. Key findings demonstrated that MTB strains overproducing L-histidine showed diminished infectivity in human macrophages and a mouse infection model, positioning metabolic activation as a promising bactericidal mechanism.

We thank the reviewer for the careful appraisal of our study and for praising its novelty and impact.

Points of discussion

1. Human Macrophage Model Analysis: the study's infection assays in human blood-derived monocytes highlighted significant differences between the allosteric mutants and the wild-type strain. Mutants, especially the A249K variant, exhibited reduced uptake and growth in macrophages. Results suggest that nutrient depletion and energy imbalance driven by L-histidine overproduction hinder MTB survival within host cells. However, the lower initial uptake raises questions about whether these mutants are inherently less invasive or if macrophages possess some recognition mechanism affecting mutant strains' entry. It is also surprising that activated Macs don't take up more (which is what they usually do). Also, the display is odd (why the bars and not simple dot-plots? They don't add any information here).

We recognize that the way the experiment was conducted may have led to some confusion. Specifically, bacteria were added to macrophages, incubated for two hours, followed by washing to remove extracellular bacteria, and then incubated for an additional two hours before plating for the four-hour time point. As a result, this time point reflects not only bacterial uptake but also early killing by macrophages. Given this, it is not unexpected that mutants with reduced overall fitness compared to wild-type show a modest decrease in CFUs under these conditions.

To clarify this in the manuscript, we have revised the text to read:

“Notably, CFUs for all mutants are significantly lower than those of the parent strain at the four-hour time point, which in our case reflects not only uptake but also early killing by macrophages.”

2. Mouse Infection Model Analysis: the mouse model findings revealed a substantial reduction in bacterial load for the A249K variant, with CFU levels dropping below the inoculum level by day 70. One critical aspect omitted is whether compensatory mechanisms in the host (e.g., immune response modifications) influence these outcomes. Further analysis could include macrophage depletion studies or histopathology to ascertain immune involvement in bacterial clearance.

We appreciate the reviewer's comment and would like to clarify that host-related aspects were not omitted arbitrarily. In studies focused on antibiotic treatment and drug discovery—such as those published in *Science* (2005, 307:223–7) and *Nature* (2000, 405:962–6)—host immunity is typically not investigated. Consistent with this precedent, our study follows the established standard in the field: demonstrating the impact of treatment or mutation through log₁₀ CFU reduction in the low-dose aerosol murine model.

Given the attenuation observed with the A279K mutant, it is reasonable to expect that bacterial nodules would not be detectable and that inflammation scores would resemble those of uninfected animals. As these experiments are not standard practice in this context, we did not perform them and therefore do not have corresponding data. Additionally, the use of formalin-fixed tissues presents technical challenges for antibody-based staining.

We fully agree with the reviewer that host immune status can influence bacterial clearance. However, this does not alter the central finding of our study: that the A279K mutant exhibits multi-log killing in C57BL/6 mice.

3. Host-Pathogen Interaction: the host section underscores nutrient depletion as a primary toxicity mechanism; however, it does not investigate into potential toxic intermediates or the host's response to L-histidine overproduction. Complementary studies exploring metabolic intermediates or cellular stress responses in macrophages would clarify the broader impact on host-pathogen dynamics. Likewise, I really would like to see the actual CFU data and not only relative changes in the mouse model. Overall, it seems like as if the modified strain is barely able to establish an infection at all. If the absolute amount at day zero is 80 CFU and on day 20 something is 160, the relevance of these findings are questionable.

We appreciate the reviewer's interest in additional metabolic data. While we agree that further exploration could be informative, our core observation is that the phenotype associated with our mutants is largely mitigated by the addition of excess carbon. This suggests that nutrient depletion—rather than broader metabolic disruption—is the dominant driver of the phenotype. Although we acknowledge that other metabolites may change over time, our data do not currently support a more complex metabolic interpretation.

In response to the reviewer's suggestion, we have replotted Figure 4B to show absolute CFU values. This allows clearer visualization of the A249K mutant's growth dynamics in mice. As shown, the mutant does grow beyond its inoculum, albeit more slowly, and becomes increasingly attenuated around day 30 post-infection. Between days 97 and 120, CFU counts range from 10 to 50—not zero. We thank the reviewer for this helpful suggestion, which improves the clarity of our data presentation.

4. Data Representation and Clarity: the CFU data presentation could benefit from a clearer breakdown of statistical variations between different time points, especially in macrophage studies. A dose-response relationship between the amount of L-histidine produced and the reduction in bacterial load would also improve interpretation. Same is true for mouse infection studies, see comment above.

We appreciate the reviewer's feedback and have added a new plot correlating CFU changes during macrophage infection with L-histidine production levels, as suggested (see new Figure S6). While this addition provides useful context, we note that correlating L-histidine levels measured in vitro after 16 hours with CFU outcomes may have limited interpretive value. Both Middlebrook 7H10 agar and macrophages—whether in vitro or in vivo—are likely exposed to some level of L-histidine, which could further attenuate our mutants. Nonetheless, we agree that presenting this correlation adds clarity and have included it for completeness.

5. No host component in the mouse model: the host component lacks an in-depth discussion of how L-histidine synthesis pathways might cross-regulate other essential bacterial pathways during host infection. Such exploration would expand on the observed fitness deficits, offering a more comprehensive perspective on bacterial metabolism and potential synergistic drug targets.

We appreciate the reviewer's suggestion and agree that we did not explore that aspect in depth. However, it remains unclear why over-activation of L-histidine biosynthesis would necessarily cross-regulate another pathway. In our study, we focused on two areas of metabolism—other amino acid biosynthesis and redox metabolism—where a direct link to L-histidine might reasonably be expected, given its role as a precursor to ergothioneine. We did not observe significant changes in these pathways. Based on our data, the observed phenotype appears to be largely explained by carbon and energy starvation, without requiring additional cross-regulatory mechanisms. In the interest of parsimony and scientific rigor, we have chosen not to speculate on further metabolic interactions that are not supported by our current evidence.